# Effect of Treated Time of Hydrothermal Etching Process on Oxide Layer Formation and Its Antibacterial Properties

**DOI:** 10.3390/biomimetics7030091

**Published:** 2022-07-07

**Authors:** Nayeon Lee, Jooyoun Park, Raheleh Miralami, Fei Yu, Nikole Skaines, Megan Armstrong, Rachel McDonald, Emily Moore, Alicia Viveros, Nicholas Borow, Keun Seok Seo

**Affiliations:** 1Center for Advanced Vehicular Systems, Mississippi State University, Starkville, MS 39759, USA; raheleh@cavs.msstate.edu; 2Department of Comparative Biomedical Sciences, College of Veterinary Medicine, Mississippi State University, Mississippi State, MS 39762, USA; jpark@cvm.msstate.edu (J.P.); seo@cvm.msstate.edu (K.S.S.); 3Department of Agricultural and Biological Engineering, Mississippi State University, Mississippi State, MS 39762, USA; fyu@abe.msstate.edu (F.Y.); nskaines2009@gmail.com (N.S.); mla349@msstate.edu (M.A.); rpm230@msstate.edu (R.M.); em1601@msstate.edu (E.M.); amv255@msstate.edu (A.V.); njb232@msstate.edu (N.B.)

**Keywords:** implant, antibacterial, nanotextured surface, biofilm

## Abstract

Inspired by natural materials, we developed an antibacterial surface on titanium (Ti) using hydrothermal etching techniques and examined the effect of treated time on oxide layer formation, its antibacterial properties, and surface defects. Hydrothermal etching was conducted on Grade 2 commercially pure Ti immersed in 5M NaOH at 250 °C during a range of time of 0–12 h. Nanopillars generated on the surface had ~100 nm thickness, which resulted in decreased attachment and rupturing of the attached bacteria. The results also showed that 6 h and 8 h of etching time provided a desirable uniform nanopillar structure with the most effective prevention of bacterial adherence on the surface. Multiscale SEM observations revealed that the longer the etching was conducted, the more cracks propagated, which led to an increase in dissociated fragments of the oxide layer. In the 12 h of etching, a higher density of bacterial adherence was observed than that of the untreated and the shorter time treated samples, indicating that etching took longer than 10 h worsened the antibacterial properties of the nano-patterned surface of Ti. This study demonstrated that the optimal time duration is 6–8 h for the oxide layer formation to maximize antibacterial activity and minimize cracking formation on the surface. For future studies, we suggest exploring many possible conditions to generate a more uniform nanopattern without structural defects to secure the integration between a newly deposited oxide layer and the substrate.

## 1. Introduction

Titanium (Ti) and Ti alloys are widely used materials as orthopedic and dental implants due to their suitable properties, such as biocompatibility, lightweight, high strength, high corrosion resistance, and lesser toxicity [1,2]. However, they frequently fail due to bacterial infection. This infection can be derived from various causes, including surgical equipment, skin, operating room, or implantation device itself [3]. Once bacterial contamination occurs, microbes adhere to the surface, duplicate, develop as microcolonies over the surface, and eventually become embedded within a self-secreted matrix of extracellular polymeric substances called biofilm [4,5]. The problem of biofilm lies on its extreme antibiotics-resistance and increased risk of recurrent/chronic infection [6]. Resistance to antimicrobial disinfectants and antibiotics is a severe issue resulting in 700,000 deaths each year worldwide. This number is expected to increase to 10 million by 2050 [7]. In the United States, up to 80% of infections are caused by biofilm, with more than 2 million cases every year, resulting in $5 billion additional medical costs [8]. During the past two decades, many research efforts have led to the development of antibiotics and coating methods with antibiotics, oxidants, or biocides. However, the infection problem is still significant [9,10].

Interestingly, it is known that living organisms ward off bacterial infection using several means along with the innate immune system. In chemical approaches, plants secrete antiseptic/antibacterial agents such as spices and oils [11,12], and animals defend themselves by secreting antimicrobial peptides and chemicals found in their skin or cuticle [13,14,15]. Another method found in nature is surface microstructures, such as insects’ cuticles, plant leaves, and mollusk shells. These are designed with nanopatterns to mechanically hinder bacterial attachment or kill microbes by perforation [16,17,18,19]. For example, microtextured structures are found in the beetle’s elytra with the shapes of polygonal walls and a diameter of ~5 μm [20,21], Cicada’s wings with bactericidal nanopillar patterns [22], and the taro leaf with water-repellent nanostructures [23] (Figure 1).

Using nanosurface to prevent infection from the bacterial attachment stage has proven to be an effective way of dealing with infection. It is more manageable to inhibit adhesion and kill the bacteria upon attachment than repel them after attachment. When the biofilm has been developed, removing it in full is very difficult [24]. The antibiotic cannot penetrate the inner part of the biofilm during antibiotic treatments, where bacteria develop antibiotic resistance [25,26,27]. Moreover, nanosurface modification can be prolonged for a long time and can be used as a powerful complement to antibiotic treatment, especially when used inside of the body, where antibiotics or non-surgical treatments can hardly reach.

Inspired by the nanosurfaces of biological materials, many attempts have been made to develop antibacterial surfaces. In order to achieve the mechano-bactericidal properties, different physical models were developed to explain how to create the most effective surfaces to target bacteria, considering the size, shape, spacing, and density of the nanopillars, and also the role of gravity and van der Waals forces in rupturing the bacterial cell wall in contact with nanopillars [16]. Furthermore, different techniques have been used to produce nanosurface, including hydrothermal synthesis and etching, plasma etching, various types of lithography (soft lithography, nano-imprint lithography, laser interference lithography, deep ultraviolet lithography, X-ray lithography, and colloidal lithography), reactive-ion etching, focused ion beam milling, 3D nano printing, oxygen plasma treatment, electrodeposition, chemical and vapor deposition, micro-molding, vacuum casting, sol-gel, etc. [6,16,22]. Of these numerous techniques, hydrothermal etching techniques are relatively economical, very effective, and the byproduct of the process, TiO^2^, is comparably biocompatible [28]. TiO^2^ is widely used for cosmetics, food products, pharmaceuticals, and biomedical applications [29]. Also, the surface of Ti and Ti alloy reacts naturally with oxygen in the air, resulting in a thin oxide film layer composed of TiO^2^. This oxide layer acts as a protective layer to make Ti highly corrosion resistant. As the etching process only adds the nanotextured pattern onto TiO^2^, the resulting material of the etching process does not contain harmful compositions. Due to the non-toxicity of Ti and TiO^2^, etching processes are highly recommended to advance the surface design of the implants.

While creating function-specific antimicrobial surfaces, the chemistry, biocompatibility, toxicity, and durability of substrates are important factors to be considered [16,30,31]. In this study, we employed hydrothermal etching techniques to create and characterize the nanotextured surface of orthopedic grade Ti. There have been several attempts to generate antibacterial nanotextured surfaces for orthopedic and dental implants using hydrothermal etching, and the previous studies report that the titanium surface produced by this technique could successfully reduce the bacterial counts [6,18,28,32]. In this study, we examined the effect of treated time of the etching on surface structure and antibacterial properties using methicillin-resistant *Staphylococcus aureus* (*S. aureus*), since *S. aureus* is the most common human pathogen causing recurrent and chronic infection in orthopedic and dental implants by forming biofilm and building antibiotic resistance. This study aimed to characterize the properties of nanotextured surfaces generated by different hydrothermal etching conditions and examine the antimicrobial properties of those nontextured surfaces against *S. aureus*.

## 2. Materials and Methods

### 2.1. Hydrothermal Etching Procedure to Generate TiO^2^ Surface

Grade 2 commercially pure Ti (cp-Ti) and Ti alloy (typically Ti-6Al-4V) exhibit similar mechanical properties, osseointegration, and biomechanical anchorage, thus both are used for orthopedic and dental implants widely [33,34]; Grade 2 cp-Ti was chosen for the substrate in this study. Cp-Ti plates (McMaster-Carr, Elmhurst, IL, USA) with a thickness of 0.3175 cm were cut into round-shape billets with a diameter of 1 cm using a waterjet machine. The billets were then machine polished using Struers Labopo-2 and TegraPol-11 polishing machines with a silicon carbide 1200/4000 grinding paper, followed by fine polishing with colloidal diamond powder (3µm). After polishing, the billets were cleansed ultrasonically, first with ethanol, and then deionized water for 15 min each to eliminate surface contaminants and particles. Each time, three of those polished and cleansed Ti billets were placed inside a Teflon liner with 100 mL of 5 M NaOH solution, then placed into a 200 mL stainless-steel reactor, which was put into a pre-heated furnace set to 250 °C for hydrothermal etching process (Figure 2). Hydrothermal etching is usually conducted using 1–10 M of alkali solution, 100–300 °C temperature, and 2–8 h, depending on the final structural designs [35,36]. In this study, to examine the effect of treated time, the authors designed the experiments with a range of treated time of 0–12 h (Table 1). Since it is impossible to observe a nanocrystal’s growing status as a function of time during the hydrothermal etching process, we set up seven different reaction times in 0 h (controlled), 2 h, 4 h, 6 h, 8 h, 10 h, and 12 h separately to determine the effect of the reaction time on the growth of the TiO^2^.

Once removed from the furnace, the reactors were placed into a cold-water bath to be cooled for ten minutes. The billets were then removed from the reactor and placed into the sonicator with deionized water for 15 min to remove any residual NaOH and particles on the surface.

### 2.2. Surface Characterization

After the etching process was done, the surfaces of the samples were examined, namely their micro- and nanostructures, using SEM analysis, elemental composition analysis, surface roughness measurement, and contact angle analysis.

SEM micrographs were taken using Zeiss SUPRA-40 field emission gun (FEG)-SEM. Three samples from each condition were fixed on stub using carbon tape and sputter-coated with a gold/palladium mixture. Then, we assessed the surface structure at a meso-, micro-, and nanoscale. The thickness of the nanopillars was measured via analyzing two-dimensional images using IMAGE J software (National Institutes of Health, Bethesda, MD, USA). The energy-dispersive X-ray spectroscopy (EDX) on the FEG-SEM was also used to carry out chemical analysis on the surface of the samples that were not sputter-coated to measure the relative amount of weight for each chemical component. 3D surface profiling was conducted using Talysurf CLI 2000 (Taylor Hobson Ltd., Leicester, UK). Each billet was analyzed at a spacing of 0.5 µm and a 1000-point resolution for a 300 µm × 300 µm area. Each case was done with the testing on five billets, and two extreme cases of the highest and the lowest values were excluded from the result data. In order to quantify the hydrophobicity, the contact angle was measured using DSA 100 to capture the image of the deionized water droplet of 0.5 μL liquid volume on the surface-treated Ti billets six times. The contact angles of the water droplets were then analyzed using ImageJ software (National Institutes of Health, Bethesda, MD, USA) with a plug-in for measuring a contact angle.

### 2.3. Bacterial Adherence Assay

Methicillin-resistant *Staphylococcus aureus* (*S. aureus*) LAC strain was obtained from Network on Antimicrobial Resistance in Staphylococcus aureus (NARSA), USA300 LAC, a well-characterized human isolate [37]. *S. aureus* was chosen because it is considered one of the leading causes of infection in orthopedic and dental implants [38,39,40]. For a stable visualization of bacterial adherence to the billets, *S. aureus* LAC strain containing chromosomally integrated green fluorescent protein (GFP) was generated by genetic modification, as previously described [41], used for bacterial adherence assay.

In order to test the anti-adherence properties of the surfaces of the treated billets, the bacteria were cultured in 5 mL brain heart infusion (BHI) broth at 37 °C overnight with shaking at 200 rpm. Bacterial suspensions were adjusted to 1×10^6^ colony forming units (CFU) per mL of BHI broth. In the meantime, all the hydrothermally etched billets were first sterilized in 70% alcohol. The sterile untreated (control) and treated billets were immersed in 1 mL of bacterial suspension in each well of the 24-well cell culture plate and incubated statically at 37 °C for 24 h. These billets were washed three times with 1 mL of PBS on a microplate shaker to remove non-adherent bacteria. The images of adherent bacteria on the billets in triplicates were collected by assessing the fluorescence of adherent bacteria using an in vivo imaging system (IVIS) (IVIS Lumina XRMS II, PerkinElmer). Each billet on the image was selected as regions of interest (ROIs) encompassing the circular billet surface, and the average of radiant efficiency [p/s/cm^2^/sr]/[μW/cm^2^] from each billet in triplicates was quantified using IVIS Image version 4 software. Then, adherent bacteria on billets were placed in 1 mL of PBS, vortexed vigorously, and enumerated by an agar plating method after 10-fold serial dilution. The statistical significance of data from different treatment groups was analyzed by Student’s t-test for simple comparisons against CT using GraphPad Prism (*p* < 0.05). Bacterial adherence assays were repeated three times. For SEM analysis, billets with adherent bacteria were fixed with 1 mL of 4% paraformaldehyde for 24 h, and the micrographs were taken using SEM.

## 3. Results and Discussion

### 3.1. Characterization of Microstructure (SEM)

Nano-patterned surfaces were created using hydrothermal etching techniques. A hydrothermal etching process is simple, low-cost, and environmentally friendly compared to many other nanostructure-generating techniques [30]. When pure Ti billets were immersed in an alkaline solution at a high temperature (250 °C), Ti first dissolved into the solution, which was later combined with Oxygen ion, growing titanium dioxide (TiO^2^) on the surface [42,43]. Deposited TiO^2^ grew nanopillars of various heights, shapes, diameters, and spacings depending on the initial conditions of temperature, pressure, and time duration [44,45,46]. After the hydrothermal etching process, the nanopillar structures on the surfaces were characterized and quantified by several assessments, including scanning electron microscopy (SEM), surface roughness evaluation, contact angle measurement, and bacterial adherence assay to evaluate the antibacterial properties of the nano-patterned structure.

In the controlled billets, which were not treated with hydrothermal etching, irregularities and defects that remained after the polishing procedure on the surface were observed (Figure 3a). The possibility that these irregularities could affect the nanopillar configurations was raised due to the random growth of nanopillars. The billets treated with the hydrothermal etching for 2 h did not show significant new features at the meso and microscale; however, at the nanoscale, it was observable that they started to grow flake-like nanostructures shown in Figure 3b. In the 4 h samples, initiation of cracks on the oxide layer were observed at the microscale. At the nanoscale, a nanopillar structure was firstly observable, indicating that 4 h was the minimum time to treat hydrothermal etching using 250 °C and 5 M NaOH solution. The nanopillars forming with diameters 100–500 nm were observed consistently after 4 h up to the 12 h reaction time without significant variation in nanostructure (Figure 3c–g). The size and shape of the nanopillar structures are consistent with other previous reports for antibacterial nanopillar structures. This size of the nanopillars is adequate to inhibit attachment or even to kill 1–10 μm of microbes [18,19]. Tang et al. [47] also fabricated nanopillars with 60 nm diameter, showing a 65.5–99.5% killing rate. Lee et al. [48] produced a nanopillar array with 570–710 nm, demonstrating that this size worked to show antibacterial effect for both Gram-negative (*E. coli*) and Gram-positive bacteria (*S.aureus*). In the 6 h samples, small cracks covered the entire oxide layer, as shown in the microscale image. From the 8 h samples to 12 h samples, the cracks became so severe that they could be observed at the mesoscale, and in the 12 h sample, the oxide layer was dissociated from the substrate.

### 3.2. Surface Topography

Another depiction of the surface is shown in Figure 4, obtained from the surface topography images and the averaged surface roughness for each reaction time from 0 h to 12 h. Considering a resolution of 0.5 μm, this analysis enabled to examine the surface modification as the etching processes carried on.

The greatest surface roughness was found in the controlled billets (2.28 ± 0.39 μm). Since 3μm powder was used at the last step of the polishing, this roughness value is plausible. The surface was evened out at 2 h billets due to the dissolving reaction, which produced the lowest surface roughness (0.80 ± 0.16 μm). In 2–6 h, the surface roughness was increased as TiO^2^ was precipitated and new nanopillar structures were generated. From 6 h to 12 h, the averaged surface roughness stayed consistent. This result implies that 6 h of the reaction time could saturate the precipitated structures, and after this point, redundant nanostructures could be generated as deep ditches in the surface topography were detected and increased at 8–12 h, as shown in Figure 4a, which was consistent with the SEM analysis showing oversaturated precipitation and crack propagation throughout the surface (Figure 3e–g).

### 3.3. Chemical Compositions (EDS)

The chemical compositions were also analyzed on the treated surface using the EDS technique. Figure 5 shows that the main constituents of the oxide layer are titanium (Ti), oxygen (O), and a small amount of sodium (Na) and potassium (P), indicating that titanium oxide (TiO2) is well generated. The small amount of Na is the residue from the NaOH solution. We also measured chemical compositions on the peeled-off region where the oxide layer was dissociated when the hydrothermal etching was treated for more than 8 h. The chemical composition of that region is pure Ti, which infers that there is no gradient between the newly generated oxide layer and substrate.

### 3.4. Wettability Test

In addition, a wettability test was conducted on the surfaces. All the treated billets with the different reaction times of hydrothermal etching showed hydrophilic properties. Overall, the hydrophilicity of the billets increased as the reaction time increased. Whereas the controlled billets’ surface was least hydrophilic, 10 h treated billets showed the greatest hydrophilicity with water contact angles of ~15° (Figure 6). It is supposed that the increased hydrophilicity of the treated billets increased the surface area of water contact, subsequently making the surface more susceptible to the water droplets. A hydrophilic nature is known to promote bacterial growth; however, it has been found that when coupled with a nanopillar nanosurface, the bacteria do attach but are killed upon attachment [49].

### 3.5. Antibacterial Effects

Lastly, to determine the antibacterial effects of the hydrothermal treated Ti surfaces after different reaction times, a bacterial adherence assay was performed using methicillin-resistant *S. aureus* (MRSA). In order to visualize the adherence of *S. aureus* on the Ti surface, an *S. aureus* LAC strain containing chromosomally integrated green fluorescent protein (GFP) was generated by genetic modification, as previously described [41], used for bacterial adherence assay. The adherence was then assessed by GFP fluorescence intensity and the quantification of adherent bacteria to each billet. As shown in Figure 7, compared to non-treated controlled billets, 6 h, 8 h, and 10 h hydrothermally treated billets exhibited significantly less adherence of *S. aureus* LAC with lower GFP fluorescence (Figure 7a,b) and CFU (Figure 7c) (*p* < 0.001). Of those cases, 6 h and 8 h treated billets were the most efficient surface to minimize bacterial adherence. Compared to the 0 h samples, which is considered as 100% of adherence, it reduced bacterial adherence to 20.7% at 6 h and 4.4% in 8 h samples. In contrast, 12 h treated billets increased bacterial adherence to a level even greater than those in the controlled billets, with 159.9% (*p* < 0.001). This analysis revealed that 6 h and 8 h treated time produced the best results with regard to minimizing bacterial adherence.

### 3.6. Problem of Cracking on the Oxide Layer

Although the results in this study demonstrated that nano-structured oxide layers are promising to inhibit bacterial adherence and thus biofilm formation, it should be acknowledged that this technique also has concerns to address to further develop reliable implant products. As our results show in Figure 7, the antibacterial function is most effective in 6 h and 8 h samples because nanopillar structures developed in 6–8 h possess optimized topologies to inhibit bacterial attachments. However, SEM micrographs also show that it started to generate cracks on the oxide layer at the 4 h sample, and as the etching continued further, at the 6 h sample, the cracks covered the entire surface in a tile-like shape, with a diameter of 50–100 μm. In the 8 h sample, the cracks started to fracture the oxide layer, resulting in the dissociation of the oxide layer from the substrate, and this disintegration of the layer became more noticeable in the 10 h and 12 h samples.

Once the oxide layer was disrupted by cracking, the antimicrobial properties were no longer effective. After culturing the bacteria in the well with Ti billets and washing, the bacteria were fixed by formaldehyde. Then, SEM micrographs were taken on the surface to observe the attachment and proliferation of bacteria. Figure 8a shows the region where the oxide layer was dissociated from the substrate at the 12 h sample. As shown in Figure 8b, bacteria were observed in the exposed substrate region. The number density of *S. aureus* in the peeled-off region is greater than that in the controlled sample, as depicted in Figure 7.

Another problem caused by the disintegration of the oxide layer is that the particles and fragments of the oxide layer can cause harmful effects inside the body. There has been a concern of metal particles falling apart in implants through wearing and corrosion-inducing inflammatory reactions. It is reported that dissolved Ti particles from implants into the surrounding bone induce inflammatory reactions in the adjacent bone, and specifically, the inflammatory cytokine tumor necrosis factor-alpha (TNF-α) is expressed [50].

Through this study, we demonstrated the effect of nanotexture generated by hydrothermal etching in reducing bacterial adherence. However, this study lacks post-processing, such as acid washing procedures or calcination processes, that could help to obtain a more controlled and well-defined crystalline phase [51,52]. In future studies, it is suggested to facilitate additional procedures to produce more reliable surface nanopillars, such as adding post-treatment or vibration-assisted hydrothermal etching.

## 4. Conclusions

Inspired by nature’s surface design found in beetle’s cuticles, dragonfly wings, and leaves, we produced surface patterns at the nanoscale using hydrothermal etching on pure Ti and examined the time effect on the oxide layer formation. The testing was performed at 250 °C using 5 M of NaOH solution at 0–12 h of a treated time with two-hour intervals. Based on structural observation and microbial testing, the results demonstrated that the hydrothermal etching with 6 h and 8 h treated time generated the most consistent nanopatterns with the lowest average surface roughness, resulting in the least amount of bacterial attachment on their surfaces. The 10–12 h samples produced nanopillar structures; however, the over-precipitation of the oxide layer caused cracking on the surface, which provided favorable sites for bacterial attachment. Thus, we conclude that the optimal condition to produce reliable surface nanostructures was 6–8 h treated time, and the nanotextured surface with well-established nanopatterns is promising in inhibiting bacterial adherence and biofilm formation in implants. For future study, we suggest investigating further to produce more reliable surface nanopillars without cracks and dissociation of the oxide layers.

## Figures and Tables

**Figure 1 biomimetics-07-00091-f001:**
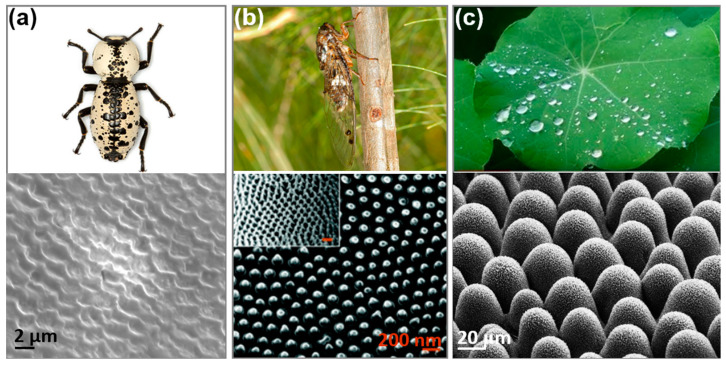
Natural materials exhibiting antibacterial/hydrophobic nano-structured surfaces: (**a**) ironclad beetle’s nanopattern on the elytron surface providing anti-adhesive properties [20]; (**b**) cicada’s wings (http://www.brisbaneinsects.com/brisbane_cicadas/FlouryBaker.htm, accessed on 2 July 2022) and its nanopillar structure to kill bacteria [22]; and (**c**) taro leaf with water-repellent nanostructure (Taro leaf image: https://www.flickr.com/photos/14731015@N06/7278281538/, SEM image: https://www.flickr.com/photos/lotus-salvinia/11081778403/in/album-72157638107887554/, accessed on 2 July 2022. SEM image © 2017, W. Barthlott, Univ. Bonn. Reprinted/adapted with permission from Ref. [23].).

**Figure 2 biomimetics-07-00091-f002:**
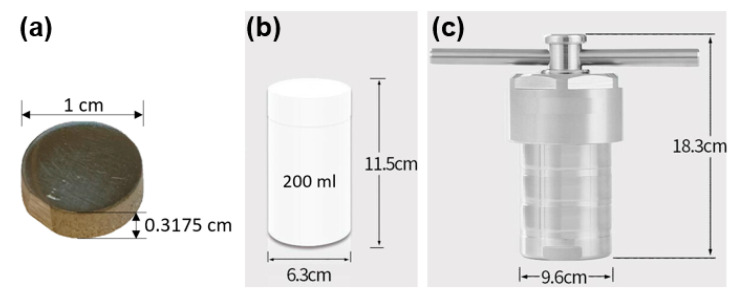
(**a**) The Grade2 commercially pure Ti billet with a 1 cm diameter round shape and a 0.315 cm thickness was treated hydrothermally at 250 °C after being placed inside a (**b**) 200 mL Teflon liner with 5M NaOH solution, which was put into (**c**) the stainless-steel reactor.

**Figure 3 biomimetics-07-00091-f003:**
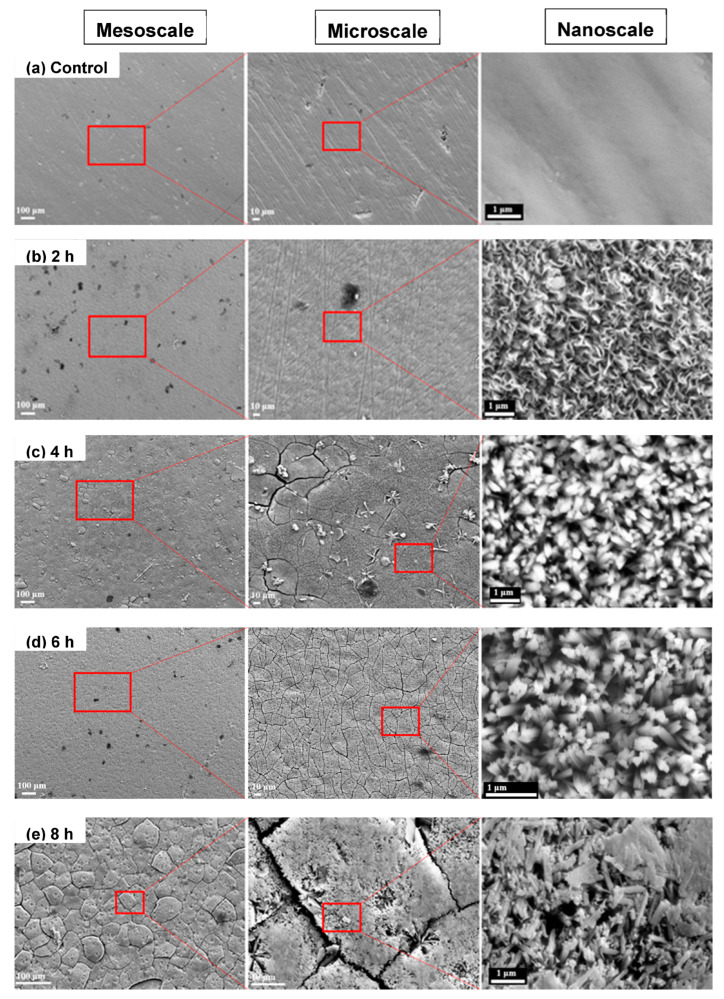
Meso-, micro-, and nanostructures of the hydrothermal treated Ti after different reaction times garnered from scanning electron microscopy: (**a**) 0 h, (**b**) 2 h, (**c**) 4 h, (**d**) 6 h, (**e**) 8 h, (**f**) 10 h, and (**g**) 12 h. At the mesoscale and microscale, it can be seen that cracking on the oxide layer has developed as the treated time increases. At the nanoscale, nanopillar structures were formed at 4–12 h treated samples with a diameter of 100–500 nm and did not show significant variations in 4–12 h.

**Figure 4 biomimetics-07-00091-f004:**
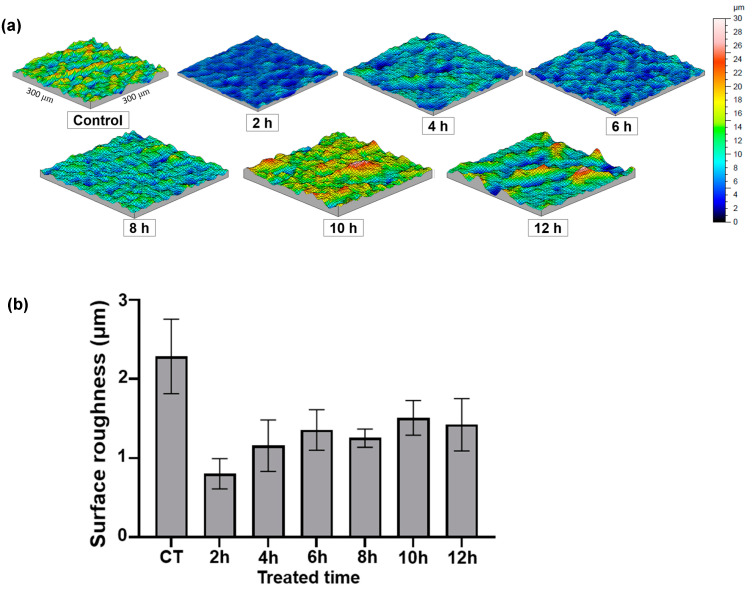
Surface topography of hydrothermal treated Ti billets after different reaction times: (**a**) surface images of scanning areas of 300 μm × 300 μm; and (**b**) averaged surface roughness (*n* = 3). The bar indicates the standard deviation.

**Figure 5 biomimetics-07-00091-f005:**
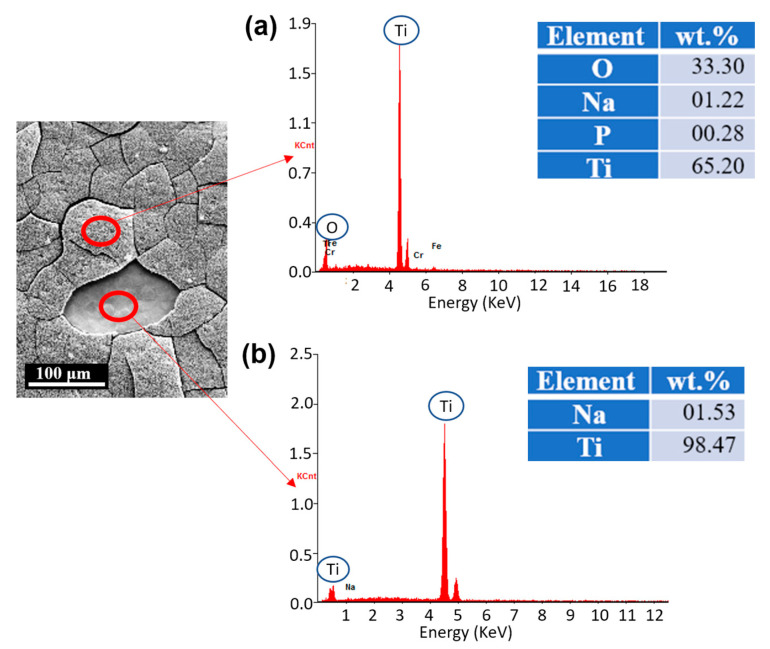
Chemical compositions of (**a**) etched surface and (**b**) the region where the oxide layer was detached. It revealed that the etched surface is composed of TiO^2^, and if the oxide layer is peeled off, then the substrate with pure Ti is exposed.

**Figure 6 biomimetics-07-00091-f006:**
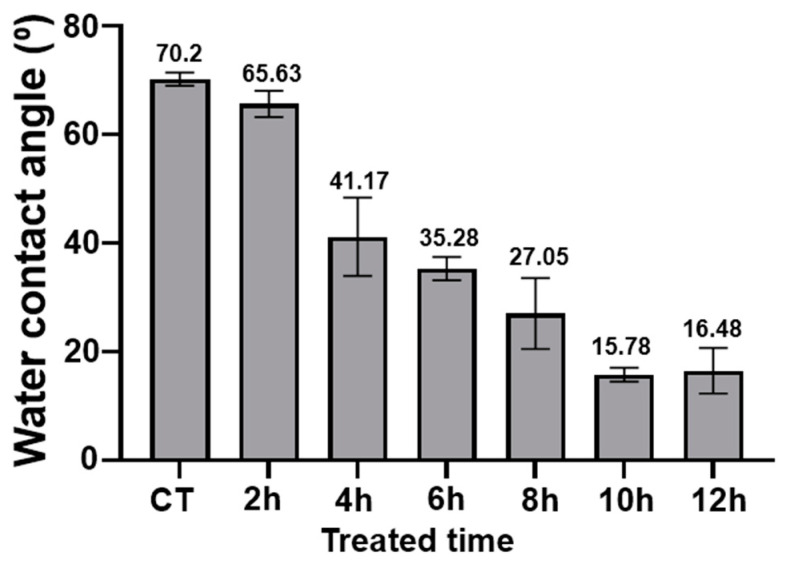
The averaged water contact angle of the hydrothermal treated Ti billets after different reaction times (*n* = 6). The bar indicates the standard deviation.

**Figure 7 biomimetics-07-00091-f007:**
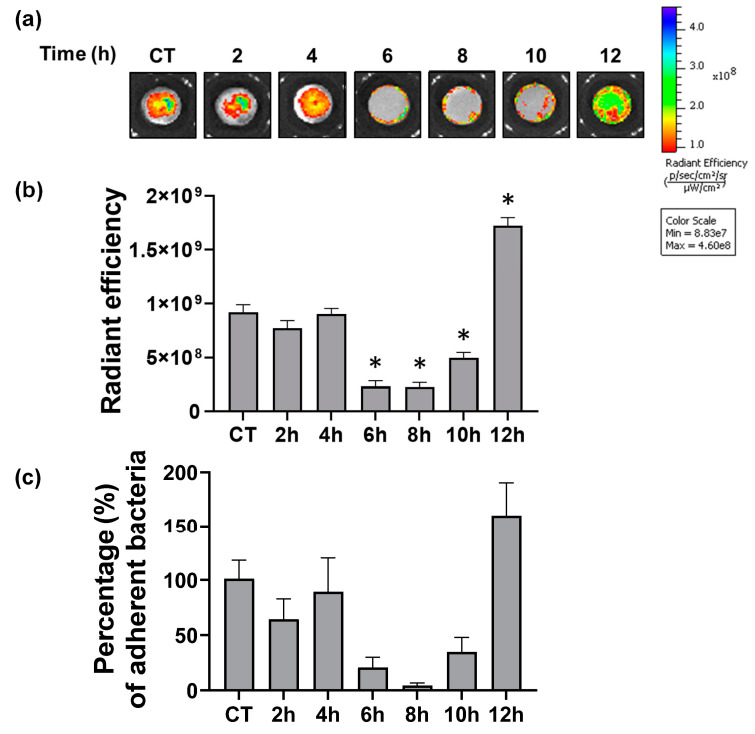
Adherence of methicillin-resistant *S. aureus* LAC on the surfaces of untreated and treated Ti with different reaction times. (**a**) IVIS images showing adherent bacteria with the intensity of GFP fluorescence. Higher intensity means more bacterial adherence to the surface. (**b**) Radiant efficiency garnered from IVIS images. A higher number indicates more bacteria. (**c**) A graph representing the percentage of adherent bacteria on the surface obtained from the CFU analysis, normalized with non-treated controlled (CT) as 100%. Data represent the mean ± standard deviation (*n* = 9). * Denoting *p* < 0.001 versus controlled sample.

**Figure 8 biomimetics-07-00091-f008:**
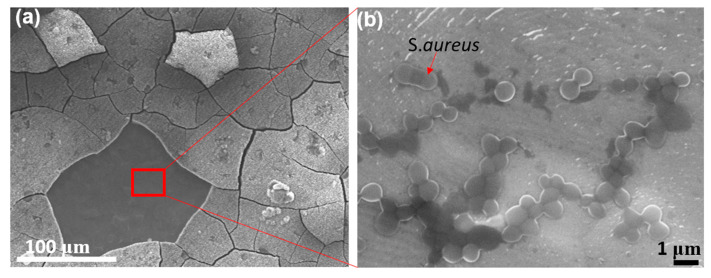
SEM micrographs on 12 h sample: (**a**) oversaturated etching causes disturbance of the oxide layer; and (**b**) the area where the oxide layer was peeled off provided a favorable surface condition for *S.aureus* to attach and grow.

**Table 1 biomimetics-07-00091-t001:** Hydrothermal etching conditions.

	Time (hours)	Molarity	Temperature
Control	0	5 Mol	250 °C
Condition 1	2		
Condition 2	4		
Condition 3	6		
Condition 4	8		
Condition 5	10		
Condition 6	12		

## Data Availability

Not applicable.

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
