# Peer review of "Effect of Treated Time of Hydrothermal Etching Process on Oxide Layer Formation and Its Antibacterial Properties"

_biomimetics, 2022, doi:10.3390/biomimetics7030091_

Round 1
Reviewer 1 Report
Overall Summary:
In this article, the authors have developed an antibacterial surface on pure titanium (Ti)
using hydrothermal etching techniques and examined the effect of treated time on oxide layer formation, its antibacterial properties, and surface defects. The authors reported that the nanopillars generated on the surface had ~100 nm thickness, which resulted in decreased attachment and rupturing of the attached bacteria. They also concluded that 6-8h of etching time at 250 ⁰C provided a desirable uniform nanopillar structure with the most effective prevention of bacterial adherence on the surface. However, with the information provided so far in the manuscript, is not enough, and some of the critical aspects is still missing. To address these deficiencies, as well as other minor ones, the authors need to consider the following concerns:
Major Concers
11. Most of the orthopedics and dental implants are made up of Ti alloys so it was not clear why the authors choose to use pure titanium for their study. They could have easily got medically grade 5 titanium for their study from the same manufacturer from which they got Grade 2 titanium. Authors are advised to provide a clear rationale for using Ti for their study.
2. Authors are also advised to provide a clear rationale for using etching time upto 12 hours?
33. The authors are advised to do a good literature review first and fixed their method section propertly. Some of the post processing steps were missing and that may be the reason the authors were seeing some surfaces of Ti and some surfaces of TiO2 on the same samples. There have been many literature already out their that have conducted similar study of creating nanotextures on Ti alloys and have tested ant-bacterial properties. One of the reference is Elliott, D. T., Wiggins, R. J., & Dua, R. (2021). Bioinspired antibacterial surface for orthopedic and dental implants. Journal of Biomedical Materials Research Part B: Applied Biomaterials, 109(7), 973-981. Authors are encouraged to do the post processing of their samples as done in this paper to fully remove NaOH and convert all the Ti into TiO2 using post calcination method. Authors are further, advised to refer this article in their introduction and discussion section too.
44. The authors mentioned creating nanopillars from their etching method, however, their SEM images were not clear at nanolevel and didn’t seem to have any nanopillars depicted. So it was not clear how they measure the length of their nanopillars. As mentioned before in the aforementioned article (Elliott, D. T., Wiggins, R. J., & Dua, R. (2021). Bioinspired antibacterial surface for orthopedic and dental implants. Journal of Biomedical Materials Research Part B: Applied Biomaterials, 109(7), 973-981.), the authors are advised to follow similar SEM protocols to get good quality SEM images where they can actually view nanopillars.
5. Authors are advised to create another section “ Statistics” in methods sections. Statistics is one of the kep information for any scientific literature and its missing here. In the Figures 3, 5 and 6, there are error bars but its not clear what they referred to , is to standard deviation, standard error ?
6. In general, authors are advised to mention the model #, manufactuer for each of their instruments that they have used. For example, in section 2.1, authors are advised to metion the details about their stainless-steel reactor ( how big was that , manufacturer, were they parr-acid digestion vessels). How many samples were put at one time?
7. In method sections, authors are advised to divide their 2.2 section into several sub sections with each tests (SEM, EDX, Contact angles, and Surface profiling) and provide details for each experiment , why particular test was conducted and what information it would give us or why its important. It is also advisable to provide the sample size and sample preparation step for each different kind of experiment in their respective section such as for surface morphology or preparation of samples for performing SEM. TWere the samples coated with gold or pd before performing SEM? In the method section, authors are advised to provide details for each of their experiment so that they can be repeated. Many key elements were missing.
8. In section 2.3, authors are advised to provide more details on the in vivo imaging system and how the images were collected using that.
9. Sample sizes have not been mentioned in most of their experimental design. For which studies, they perform the statistics? The authors are advised to provide statistical analysis section in the method section and provide some analysis in the discussion section.
10. It is not clear how the authors got the nanopillas in the range of 100-500 µm as mentioned in their results and discussion section on page 5. Authors are advised to provide clearly in the method section how these measurement were done and if possible depict the nanopillars in the SEM images for which measurement were done?
11. It is also not clear if the peeling off the oxide layer was done intentionally or it happens on its own (Figure 4). Authors are advised to provide clear and brief description of that.
12. For Figure 6 (d), I don’t completely agree with the authors inference of the image. From the current image, it may seems that is out of focus for the bacteria or the formation of biofilm what the authors has labelled ruptured bacterium. Also, it is still unclear how the authors reached to the conclusion about the alive and ruputured bacterium from the SEM image. It’s very difficult to infer that from the given SEM image. Authors are advised to provide a justification or re-write this section to be consistent with the images they got. Another possibility is that they can go to higher magnification and see the nanopillars as show in one of the already published paper by Elliott et al.
13. For Fig 7 (b and c), the authors are advised to provide the clear images.
14. Figures 2, 3(a) 4, 6 (d), 7(a, b and c) shows single images from each test group and the positive control. Are these images representative from multiple sampling throughout each group? If so, then what is needed for these data to be fully accepted would be for the investigators to quantify data in numerous images from each group. This analysis would provide objective and quantifiable outcomes that would properly inform the reader of the surface chareacteristic and bacterial viability in each test group.
Minor Comments
15. Section 2.3, there were typos for cell counts 1X106 should be replaced by 1X 10^6. Similarly, radiant efficiency unit cm2 should be replaced with cm^2
16. Page 5, in the results and discussion section, 5M acid solution should be replaced with 5M NaOH solution.
17. Authors are advised to clearly write in the Figures caption for 3, 5 and 6 about what the error bar represents and represent in the figures if they find any statistics significance?
18. For Figure, 6 (c), authors are advised to provide a clear Y-Axis title (May be percentage of adherent cells instead of just percentage)
19. The last paragraph of the results and discussion section has an abrut ending. Authors are advised to re-write the sentence to provide a clear message.
Reviewer 2 Report
Overview and general recommendation:
In sum up, this is a well-written manuscript. I recommend its publication, but some major revisions should be addressed before reconsideration. Several suggestions have been given below:
Figures and captions should be placed on the same page as much as possible.
Reviewer 3 Report
While some simple experimental procedures were done in this study, there are also some good analytical results and discussions. Therefore, some minor modifications are recommended-
Abstract: The abstract should specify, for example, the grade of titanium used, the etchant, and the hydrothermal conditions studied. I suggest modifying this section.
Introduction-lines 79-81: Expression needs to be corrected, numbers should be subscripted.
Mater and Method- section 2.1.- Please mention the reactor size, possibly an image is preferred.
Section 2.3.- line 147- There are many mistakes in expression, please check carefully through the manuscript
Results and discussions- Lines 176-182- This paragraph should be moved to the Materials and methods section
Line 187- Please unify unit expressions, such as hr vs h.
Figure 3- Please add statistical comparison results and provide n values for the number of repetitions in all Figures needed.
Author contributions- This part should be specifically modified.
Round 2
Reviewer 1 Report
Well, most of the concerned have been addressed appropriately by the authors, however, there are still few left that still needs to be addressed further.
1. Previous comment # 4. I understand due to time constraints, authors cant perform the calcination step now. Authors are advised to mention that they didn’t perform the calcination process in the discission section as one of the drawbacks of their study.
2. SEM images in Figure 3 at high resolution (Nanoscale) are still blurred, authors are advised to provide a better quality of images if possible.
3. Previous comment # 12: It seems to be an over interpretation of the Figure 7 (d) and Figure 8 (b). It is not clear how the authors are reaching to the conclusion that its even a dead bacterium. What the authors labelled as dead bacterium can be the bio film or 3-4 bacteria over its each other in a different plane getting out of focus giving them a blurred picture. Authors are advised to fix the images (even the labelling) and do appropriate discussion in the discussion section without overstating.
